# Analysis of Multiple Risk Factors for Seronegative Rate of Anti-Tick-Borne Encephalitis Virus Immunization in Human Serum

**DOI:** 10.3390/medicina56050244

**Published:** 2020-05-20

**Authors:** Marta Janik, Sylwia Płaczkowska, Mieczysław Woźniak, Iwona Bil-Lula

**Affiliations:** 1EUROIMMUN POLSKA Sp. z o.o., Widna Str. 2a, 50-543 Wroclaw, Poland; m.janik@euroimmun.pl; 2Department of Laboratory Diagnostics, Diagnostics Laboratory for Teaching and Research, Wroclaw Medical University, Borowska Str. 211a, 50-556 Wroclaw, Poland; sylwia.placzkowska@umed.wroc.pl; 3Department of Medical Laboratory Diagnostics, Division of Clinical Chemistry and Laboratory Hematology, Wroclaw Medical University, Borowska Str. 211a, 50-556 Wroclaw, Poland; mieczyslaw.wozniak@umed.wroc.pl

**Keywords:** diagnostics, vaccination, tick-borne encephalitis virus, anti-Tick-borne encephalitis virus antibodies, waning immunity

## Abstract

*Background and objectives:* Tick-borne encephalitis virus (TBEV) infections have been the cause of threatening outbreaks for many years. Apart from several physical and chemical methods to prevent tick bites, active vaccination of people highly exposed to infection is still the most important strategy of prevention. However, in some subjects, the lack of or low response to TBEV antigens is observed. The aim of the current study was to assess the prevalence of seronegative rate for anti-TBEV antibodies and the risk factors for waning immunity. *Materials and Methods:* 2315 at least primary vaccinated subjects from the high risk group for TBEV infections participated in this study. A commercial enzyme-linked immunosorbent assay (ELISA) test was used for the assessment of anti-TBEV IgG serum level. *Results:* Data showed that 86.2% of subjects who underwent vaccination were positive for anti-TBEV antibodies within 5 years. As much as 13.8% of subjects that underwent primary or primary and booster vaccination were barely protected after vaccination. Women and subjects under 60 years underwent more effective protection but sex and older age was not a risk factor for being a subject of waning immunity. A logistic regression showed that both a longer time since the vaccination and a lower number of booster doses constantly increased the chance of lost anti-TBEV antibodies. *Conclusions:* This study demonstrates that the vaccination schedule should be reevaluated. The extension of the interval of booster immunization is risky and all subjects should be surrounded by care consisting of more frequent monitoring of serum antibodies by personalized schedule to adjust the frequency of subsequent doses of booster vaccination.

## 1. Introduction

Tick-borne encephalitis (TBE) is one of the most serious human neuroinfections that is endemic in areas of Europe and Asia, beyond borreliosis, erlichiosis, babesjosis and Q fever [1]. There are more than 10 thousand severe Tick-borne encephalitis virus (TBEV) infections of the central nervous system (CNS) in Europe per year [2] and a permanently large increase (140%) of tick-borne encephalitis has been reported between 1991 and 2010 [3].

The etiological agent of TBE is a Tick-borne encephalitis virus, a member of the genus Flavivirus and family Flaviviridae, which is transferred to humans by Ixodes ticks [1]. The infection usually occurs through a prick of the tick, but there are also cases of virus transmission through the gastrointestinal tract, the consumption of non-pasteurized milk and milk products or through the respiratory tract [4,5]. The TBE infection may lead to meningitis, encephalitis, myelitis, central nervous system injury and fatal outcome [1,5,6]. The severity of clinical manifestation is associated with the age and some comorbidities such as diabetes mellitus and hypertension [6,7].

The diagnosis of TBE is usually based on the clinical manifestation and declaration of tick bite. Unfortunately, only 50–60% of patients can remember the tick bite incident. Moreover, due to the low specificity of clinical manifestations and a similar course with other CNS infections, the diagnosis of Tick-borne encephalitis has to be confirmed by laboratory tests of blood and cerebrospinal fluid (CSF) [8,9,10,11]. Although the assessment of anti-TBEV immunoglobulin G (IgG) in either blood or CSF during the second phase of the disease is the basis of routine diagnosis of TBE, there are some limitations that may cause diagnostic problems, like persistent immunoglobulin M (IgM) antibodies in serum, and cross-reactivity of the antibodies [5,7,8,10,11,12,13,14].

Nowadays, the active vaccination of people highly exposed to infection (for example employees of forest institutions or people living in or visiting endemic areas) is the most important strategy of prevention that limits a morbidity and mortality due to TBE [15,16]. It has been documented that active immunization results in a high rate of seroconversion and led to an extremely low number of TBE cases [9,17]. Simultaneously, data from the other studies indicate a significantly lower degree of seropositive ratio after TBEV vaccination and its dependence on age and the number of booster doses [18,19]

Since 1976, FSME-Immun^®^ (Baxter, Wien, Austria) and Encepur^®^ (Novartis Vaccines, Marburg, Germany) are commonly used in the Europe. The conventional schedule of the basic vaccination course consists of 3 doses, administered in the interval of 1–3 months (between first and second dose) and 5–12 months (between the second and the third dose). Accelerated schedules can be applied in emergency situations (vaccination on days 0 and 14, followed by a third dose 5–12 months after the second). The basic 3-dose schedule may be followed by booster doses every 3–5 years (once every five years in subjects under 60 years old and once every 3 years in subjects over 60 years old) [11,17]. Seroconversion is considered to have been achieved when a vaccine has a negative neutralization test (NT) (<1:10) or positive results of Enzyme-Linked Immunosorbent Assay (ELISA) (the titer cut-off depends on the ELISA kit manufacturer) before the first [16].

The scheme of vaccination protocol is shown in Figure 1, and may vary from country to county. Since basic vaccination does not provide the full immunization, several booster doses are recommended [17].

The utility of certain vaccines is limited for some reasons. Firstly, adverse vaccine-reactions has been observed [20]. Secondly, there are two types of vaccination failure called “non-responders” (any humoral response to vaccination is observed 4–6 weeks after immunization) [17,21] and waning immunity when antibodies against vaccine’s antigens are produced but were not able to maintain the immunity they previously produced [22]. For these reasons it is highly important to control immunological status after vaccination, which allows for verification of vaccination efficiency and the presence of proper humoral response to the virus. So far, the problem of persistence of immunization after TBEV vaccination has not yet been widely studied in Poland.

The lack of specific treatment of TBE draws attention to the necessity of nonspecific prevention such as personal protective procedures and the reduction of tick population [23,24], but an especially important issue is the possibility of passive immunization by vaccination [25]. Therefore, the aim of the current study was to assess the presence and the level of anti-TBEV antibodies in serum and the time course of the antibodies in vaccinated subjects. The specific aims were to identify the subject of waning immunity and to evaluate the risk factors leading to insufficient and nonprotective anti-TBEV antibodies level, considered as titer under seropositive level.

## 2. Materials and Methods

### 2.1. Study Group and Clinical Material

All participants were recruited by the Euroimmun Polska Sp. z o.o. Company (Wroclaw, Poland). Vaccination data were acquired from a written questionnaire due to participants’ memory. All subjects were informed about the aim of the study and gave written consent for their participation. The study protocol was approved by the Bioethics Committee of Wroclaw Medical University (Poland), acceptance no. 586/2012 (date: 06/28/2012). Written informed consent was obtained for the collection of blood samples from each participant. Clinical samples of whole blood (to obtain serum) were collected. The samples were centrifuged at 3500× *g* for 7 min. Serum aliquots were stored at −20 °C before analysis in a commercial laboratory.

### 2.2. Quantitative Assessment of Anti-TBEV IgG Antibodies in Serum Samples

Commercial Anti-TBE Virus ELISA “Vienna” (IgG) test from EUROIMMUN AG (Lubeck, Germany) was used for quantitative detection of specific IgG in serum samples from study participants, according to the manufacturer’s instruction. Appropriate positive and negative control sera were used in each assay. Each sample was tested in duplicate. Results were shown in VIEU/mL.

### 2.3. Definitions and Criteria for Classification

Primary immunization is a vaccination carried out in accordance with the standard protocol of three doses of vaccination (day 0, 1–3 month after 1st dose, 5–12 months after 2nd dose). The primary immunization term used in this study means that the subject received three doses of primary vaccination in a timely manner.

Booster immunization means supplementary doses of the vaccine administered as a re-exposure to the immunizing antigen (remaining doses).

### 2.4. Criteria for Classification

The manufacturer of the FSME Vienna IgG assay (Euroimmun, Lübeck, Germany) defines the results as negative for values of <120 VIEU/mL, indeterminate for values between 120 VIEU/mL and 165 VIEU/mL, and positive for values of 165 VIEU/mL or more (FSME Vienna IgG assay (EI 2661-9601-9G; Euroimmun, Lübeck, Germany)

Since only a neutralizing test is able to assess immune response to vaccination, waning immunity was considered those with negative or indeterminate results for anti-TBEV antibodies in this study (without evident seroconversion).

### 2.5. Statistical Analysis

Statistica 12 PL software (StatSoft, Krakow, Poland) was used for data analysis. Results were expressed as a geometric mean titer with two sided 95% Confidence Interval (CI). ANOVA Kruskal-Wallis, followed by a Dunn post-hoc test were used for multiple comparison of titer in age groups. To confirm the homogeneity of compared groups depending on the time since the last dose of vaccine Chi square test has been used and then for the same groups, a Chi-squared test for trend was used to confirm a linear trend for an increasing number of seronegative cases across the time since last vaccination. To determine the predictors for a lack of vaccine-induced immunity, logistic and Cox regression analyses were used and a *p* value of <0.05 was considered to be significant.

## 3. Results

### 3.1. Prevalence of Waning Immunity in Study Group

The median age of the study group was 46.2 year (95% CI: 20–68) and was similar to the median age in the waning immunity group 47.8 (95% CI: 22–66). In all study groups, 13.78% (*n* = 319) participants presented a TBE antibodies titer below 165 VIEU/mL and were recognized as seronegative. The group with only a primary immunization seronegative rate was 24.28% (*n* = 152), since among participants subjected to primary and booster immunization it was 9.89% (*n* = 167). Characteristics of vaccination status of all and waning immunity study groups are presented in Table 1.

### 3.2. Changes of Anti-TBEV Titer in Participants Subjected to Complete Primary Vaccination Only

The analysis of the time dependent frequency of the seronegative rate of anti-TBEV antibodies concentration (<165 VIEU/mL) showed that time elapsed since the last, third dose of basic vaccination was associated with an increased number of negative results for anti-TBEV antibodies (Chi^2^ for trend, *p* = 0.012) (Figure 2).

Data showed that the serum titer of anti-TBEV antibodies decreased during the time since the last, third dose of primary vaccination. In 27.3% of subjects vaccinated ≥ 4 years before testing and as much as 14.3% of participants underwent vaccination less than 1 year before testing, anti-TBEV titer did not reach 165 VIEU/mL and they served as those with waning immunity. Of subjects subjected to basic protocol of vaccination, 85.7% reached a positive titer (>165 VIEU/mL) of antibodies within 1 year after vaccination, and 72.7% within 4 years since the last dose of primary vaccination (Table 2).

### 3.3. Changes of Anti-TBEV Titer in Participants Subjected to Complete Primary Vaccination and at Least One Booster Dose during the Time

Taking into account that the protocol for a booster immunization is dependent on subjects’ age, the analysis of subjects underwent 3 doses of the primary vaccine and at least 1 dose of booster vaccine (≥4 doses in total) were examined in two age groups (Table 3).

As much as 7.5% of subjects under 60 years vaccinated < 1 year before testing and 9.0% of subjects vaccinated ≥ 4 years before testing had negative or indeterminate titer (<165 VIEU/mL) of anti-TBEV antibodies. Of the participants, 91.0–92.5% were positive for anti-TBEV antibodies in the period of 1 to ≥4 years since the vaccination.

In a study group over 60 years old, a negative or indeterminate antibodies titer was observed in 10% of subjects less than 1 year after last vaccination. With the passing of years, the number of subjects who were seronegative reached 14.3%. Chi-squared test for the trend did not confirm the increase of the prevalence of seronegative participants increased during the time since the vaccination (*p* > 0.05) in this group.

### 3.4. An Influence of Sex and Age on Anti-TBEV Antibodies Titer in Serum

To determine whether sex and age were associated with antiviral antibodies status, an anti-TBEV antibodies titer in the serum of the participants undergoing primary or primary and booster vaccination was analyzed. We showed that an average of anti-TBEV antibodies in women was statistically higher (373, 95% CI 269–518 VIEU/mL) than in men (314, 95% CI 278–355 VIEU/mL), *p* < 0.05. However, sex was not associated with antibodies titer in subjects who received the primary and remaining vaccination (≥ 4 doses) (461, 95% CI 395–538 VIEU/mL vs. 485, 95% CI 460–510 VIEU/mL) as well as it was not associated with increased incidence of seronegative titer of anti-TBEV antibodies (*n* = 145, 9.7% vs. *n* = 22, 11.5%), *p* > 0.05 in this group.

Similar to above, the subjects’ age was evaluated. In both, subjects underwent primary vaccination as well as those with primary and remaining vaccination (≥4 of doses), the highest titer of anti-TBEV antibodies was observed in subjects between 31–40 years (464, 95% CI 389–554 VIEU/mL and 571, 95% CI 517–631 VIEU/mL, respectively). In both groups, antibodies titer decreases in older groups reaching as low a value as 207 (95% CI 130–330 VIEU/mL) and 431 (95% CI 373–498 VIEU/mL) in subjects over 60 years, respectively (Figure 3A,B), *p* < 0.01.

### 3.5. Analysis of Predictors of Seronegative Ratio

Age > 60 years, female sex and time since the last dose were evaluated as independent predictors of waning immunity in multivariate logistic regression analysis. As expected, the model built for time from the last dose of vaccine, adjusted for age <60 and ≥60 years and sex revealed a significant increase of anti-TBEV seronegative ratio after 2, 3 and 4 (and more) years (Table 4).

Booster doses of the TBEV vaccine were administered according to the vaccination schedule at appropriate time intervals, so we have used the proportional hazard Cox regression to analyze the impact of number of booster doses adjusted to age and sex. An incidence of seronegative cases negatively correlated with the number of booster doses. Of the subjects who received ≥4 periodic booster doses, 6.7% did not reach a value ≥ 165 VIEU/mL in comparison to 24.3% of subjects who received only complete primary vaccination and were tested within 1 to ≥4 years since vaccination. For Cox regression analysis following variables were evaluated: time from the last, third dose of primary vaccination, age group and sex. Results of this analysis revealed that the relative risk of seronegative rate is significantly decreased after the first booster dose (risk reduction by 86%) and this effect deepened with each subsequent dose. Moreover, in the age group < 60 years the risk of seronegative ratio was reduced by 33%, when relative risk for sex was not significant and has been excluded in forward stepwise selection of the regression model (Table 5). As hypothesized, periodic booster vaccination and age < 60 years decreased the chance for low level of anti-TBEV antibodies in the serum.

Figure 4 shows the probability of seronegative ratio for men ≥60 years depending on the number of received booster doses.

## 4. Discussion

Several previous studies suggested that some subjects who received vaccination did not respond to vaccination [17,26] and some congenital disorders, which affect antigen presentation for lymphocyte B, might be a main cause of vaccination failure [27]. Finally, severe cases of TBE have also been observed in patients previously vaccinated with a proper initial response to vaccination [21]. For this reason, non-response or waning immunity are very important and problematic clinical issues. Those patients are not protected after primary infection or vaccination and lower titers of neutralizing antibodies delay the clearance of the virus and may result in the infection of neuronal cells [28].

In the current study we showed that as much as 13.8% of all participants underwent primary or primary and remaining vaccination < 1 up to 5 years before testing did not reach a titer of 165 VIEU/mL (indicating seroconversion) and the prevalence of waning immunity was increased simultaneously to time since the last dose of primary vaccination. As much as 75.7% of subjects who underwent basic protocol were anti-TBEV positive in the ELISA test within 1–5 years. An incidence of waning immunity (negative or intermediate test results) in the group of subjects subjected to primary vaccination followed by periodic booster doses was 7.21% within 5 years. Data published by Plotkin et al. suggested that 14 days after the second dose of primary vaccination, the protective level of antibodies developed in about 85% of the subjects, and after 3 doses in about 98% of participants [29]. The most of the TBE epidemiology data in Europe comes from the three countries of its central part—Czech Republic, Slovakia and Austria—because they have had a well-established system of documentation of TBE based on TBEV IgM and IgG ELISA results. Data collected in this system was used by Heinz et al. in 2013 to calculate the Field Effectives of Vaccination in Austria, which was assessed on 96–99% of people vaccinated according to the recommended program and about 90% of those who underwent only the primary protocol [30]. Our results indicated that a quarter of persons subjected only to the primary protocol have insufficient levels of TBEV antibodies which is much higher in comparison to Heinz et al. In contrast, Baldovin et al. (2012) conducting a study in a highly endemic region of Italy have revealed that the persistence of the protective level of TBEV antibodies decline remarkably to 50% at 50 months of follow-up. When they analyzed the results in age groups for the older participants aged over 60 years, TBEV antibody levels reached 60% at 60 months and only 20% at 70 months of follow-up [19].

In our study, a positive level of antibodies (≥165 VIEU/mL) was observed on average in about 86.2% of the analyzed population over a period of 5 years since the last dose. This suggests that the waning immunity phenomenon is still observed in clinical practice. Heinz et al. (2007) showed that after complete active vaccination, antibody concentrations decreased in a log-linear way falling below protective levels within 2–4 years in persons over 60 years [17]. We showed that as much as 11.1% of subjects subjected to primary vaccination and 8.5% of those with booster vaccination were seronegative just after one year since vaccination. Taking into account that following the standard protocol this group should be vaccinated in the next three years, it means that during this time they are as susceptible as those who were not vaccinated at all.

To assess the factors affecting the humoral response to vaccination, the logistic regression analysis of predictors of non-responders were conducted. Although this study showed that anti-TBEV antibodies titer decreased with age, age was not a risk factor for being a non-responder. We showed that the highest values of anti-TBEV were observed in young subjects and the titer decreased with age reaching the lowest values in those over 60 years old. This may be related to the natural aging of the immune system which is not as effective as in younger people. There are many previous reports showing reduced productivity of the immune system in older people, which results in greater susceptibility to infections in comparison to their younger counterparts [31,32]. We also showed that as much as 7.5% of subjects < 60 years and 10% of those over 60 years were negative for anti-TBEV antibodies during the first year post-vaccination. This means that following the standard protocol of vaccination they should receive the first booster dose in 5 and 3 years, respectively, so they are not being protected during this time.

Our results also showed that women underwent much more effective protection than men. The average of anti-TBEV serum antibodies titer was higher in women than in men. However, sex was not a risk factor for being a non-responder in the whole group (those with primary vaccination as well as those with primary and booster vaccination).

There are some discrepancies that prompt us to undertake this study. In 2012 Askling et al. showed that the lack of immune response to anti-TBEV vaccination was not associated with the number of vaccine doses nor the maintaining of established time in the protocol between following doses [33]. They reported that the prevalence of non-responders was similar in groups vaccinated according to the recommended protocol as well as in groups receiving a different number of doses [30]. Moreover, Wagner et al. [16] suggested that post-vaccination serologic testing may only be reasonable in the case of unclear immunization status due to incomplete or lack of documentation of vaccination courses. In response to all the above, a logistic regression analysis of variables which might increase the risk of improper immune response to anti-TBEV vaccination was performed in our study. We showed that both longer times since vaccination and a lower number of vaccine doses increase the chance for a negative value of anti-TBEV antibodies in tested subjects. This highlights the necessity of more frequent examination of vaccinated people, who may then be classified to earlier booster immunization to reach a positive value of anti-TBEV antibodies.

Due to the lack of effective and safe strategies for the prevention and treatment of TBE, vaccinations remain an essential tool in the fight against this disease. The undoubted advantage of the TBE vaccine is the possibility of widespread use in all age groups as it is practiced in many European and world countries [9]. Although primary vaccination alone provides a significant degree of immunization, it decreases with the passage of time from the last vaccine, even when the vaccination schedule does not require a booster dose. To maintain immunity against TBEV, it is important to capture the moment when the antibody titer drops to a level that does not provide adequate protection. This situation can be detected by performing a simple and inexpensive antibody titer test. Therefore we suggest that all TBE immunized subjects, and especially those from high risk groups should be surrounded by care consisting of more frequent monitoring of serum antibody titers, and more frequent vaccination if needed.

## 5. Conclusions

In conclusion, this study showed that time schemes for booster vaccinations should be reevaluated. We showed that the extension of the interval of booster vaccinations is risky for all subjects, regardless of sex and age. We strongly suggest that all subjects from high risk groups should be surrounded by individual care, consisting of more frequent monitoring of serum antibody titers based on personalized schedule to adjust the frequency of subsequent doses of booster vaccination.

### Limitation of the Study

In the current study, ELISA for TBEV-specific IgG was used to test the immune response following immunization. We are aware that the major obstacle to using ELISA is the existence of cross-reactive, non-neutralizing antibodies induced by other flaviviruses. However, tested subjects did not declare a vaccination history against yellow fever (YF) or Japanese encephalitis (JE) nor travel history to tropical and subtropical countries. Although a neutralization test is a standard method to determine the immunization status, ELISA tests are inexpensive and widely available in clinical laboratories.

The next limitation of our study is definitely more male representation. A higher male number of vaccinated participants comes from occupational exposure. Study results obtained from female group (see Appendix A) correspond to results obtained from all study participants.

## Figures and Tables

**Figure 1 medicina-56-00244-f001:**
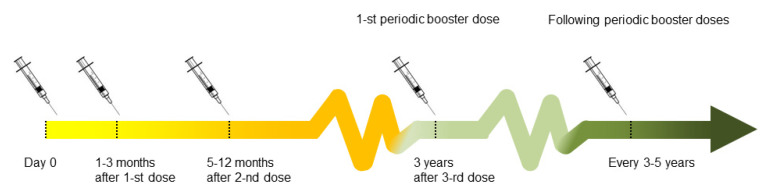
The scheme of vaccination in Austria.

**Figure 2 medicina-56-00244-f002:**
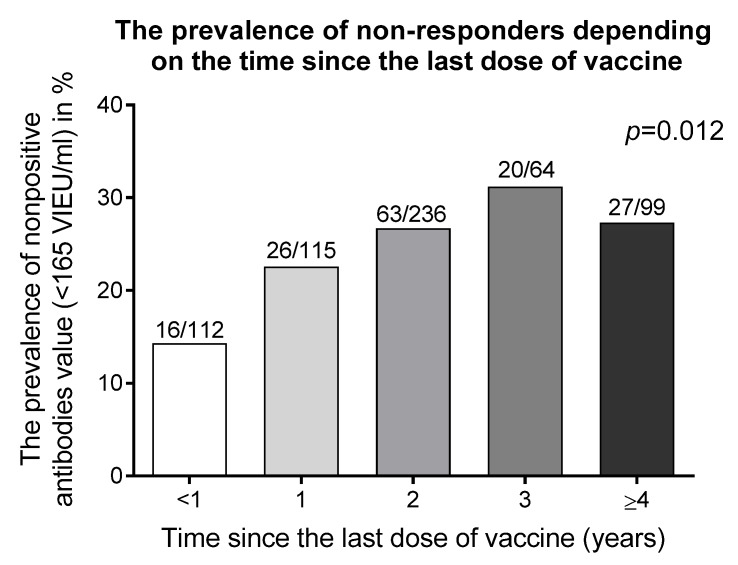
An incidence of seronegative rate of anti- Tick-borne encephalitis virus (TBEV) antibodies (<165 VIEU/mL) after primary vaccination depending on the time since the last dose. VIEU/mL—a unit of serum antibodies titer.

**Figure 3 medicina-56-00244-f003:**
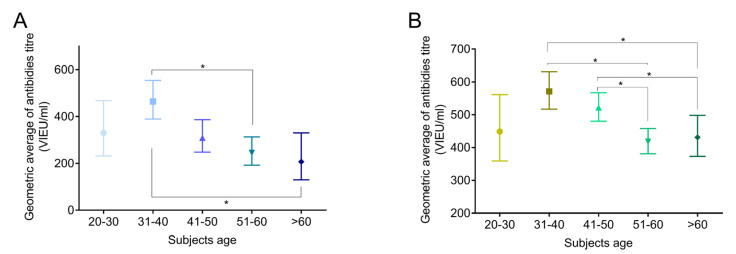
Changes of anti-TBEV antibodies in serum of subjects underwent complete primary vaccination (**A**) and subjects with primary and at least one booster dose of the vaccine (**B**) in different age groups. * *p* < 0.05. TBEV—Tick born encephalitis virus; Data are presented as geometric mean of antibody titer (VIEU/mL) with 95% confidence intervals.

**Figure 4 medicina-56-00244-f004:**
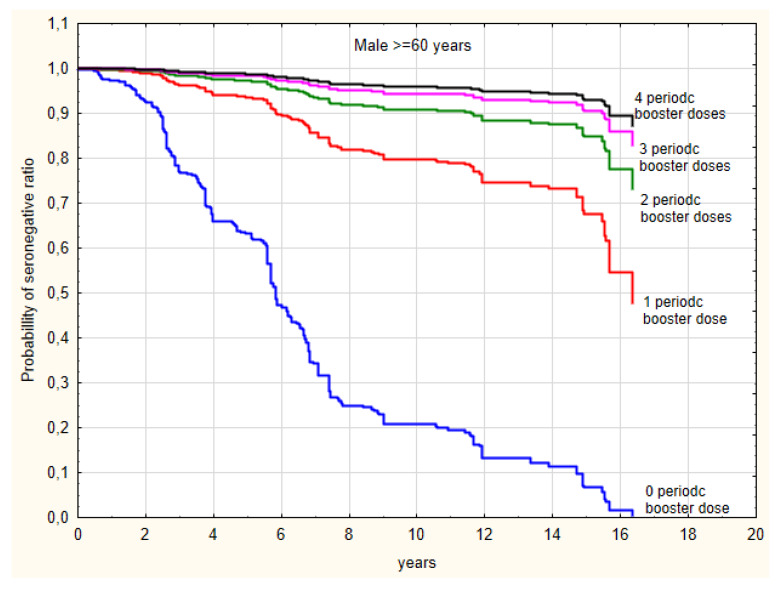
The probability of seronegative ratio for men over ≥60 years depending on the number of received booster doses.

**Table 1 medicina-56-00244-t001:** Characteristics of vaccinations status in all and seronegative study groups.

Basic Characteristics	All Study Group *n* (% All Study Group)	Waning Immunity *n* (% All Study Group)
*n* (%)	2315 (100)	319 (13.78)
Male, *n* (%)	2031 (87.7)	281 (12.14)
Primary immunization, n(%)	626 (27.1)	152 (6.56)
Primary immunization and 1 periodic booster dose, *n* (%)	991 (42.8)	118 (5.10)
Primary immunization and 2 periodic booster doses, *n* (%)	408 (17.6)	30 (1.29)
Primary immunization and 3 periodic booster doses, *n* (%)	155 (6.7)	10 (0.43)
Primary immunization and ≥4 periodic booster doses, *n* (%)	135 (5.8)	9 (0.39)

**Table 2 medicina-56-00244-t002:** Changes of anti-TBEV titer in participants subjected to complete primary vaccination only.

Anti-TBEV Titer VIEU/mL)	Time Since the Vaccination (Years)				
	Time Period Complies with the Vaccination Protocol before First Booster Immunization				Time Period beyond the Vaccination Protocol
	<1	1	2	3	≥4
	Total *n* = 112	Total *n* = 115	Total *n* = 236	Total *n* = 64	Total *n* = 99
	*n* (%)	*n* (%)	*n* (%)	*n* (%)	*n* (%)
<120	12 (10.7)	22 (19.1)	49 (20.8)	15 (23.4)	24 (24.3)
120–164	4 (3.6)	4 (3.5)	14 (5.9)	5 (7.8)	3 (3.0)
≥165	96 (85.7)	89 (77.4)	173 (73.3)	44 (68.8)	72 (72.7)

**Table 3 medicina-56-00244-t003:** Changes of anti-TBEV titer in participants subjected to complete primary vaccination and at least one booster dose.

Age of Participants (in years)	Time since the Vaccination (in years)	Antibodies Titer in Serum (VIEU/mL)		
		<120	120–164	165–1000
<60	<1 Total *n* = 401 (%)	22 (5.5)	8 (2.0)	371 (92.5)
	1 Total *n* = 321 (%)	17 (5.3)	6 (1.9)	298 (92.8)
	2 Total *n* = 480 (%)	42 (8.8)	25 (5.2)	413 (86.0)
	3 Total *n* = 156 (%)	6 (3.8)	7 (4.5)	143 (91.7)
	>4 Total *n* = 133 (%)	9 (6.8)	3 (2.2)	121 (91.0)
≥60	<1 Total *n* = 50 (%)	5 (10.0)	0 (0.0)	45 (90.0)
	1 Total *n* = 43 (%)	2 (4.6)	2 (4.6)	39 (90.8)
	2 Total *n* = 63 (%)	3 (4.8)	4 (6.3)	56 (88.9)
	3 Total *n* = 21 (%)	1 (4.8)	2 (9.5)	18 (85.7)
	≥4 Total *n* = 21 (%)	3 (14.3)	0 (0.0)	18 (85.7)

Notes: VIEU/mL—the unit of serum antibodies titer.

**Table 4 medicina-56-00244-t004:** Logistic regression analysis of risk factors related to seronegative values of anti-TBEV serum antibodies.

Time since the Vaccination in years	Anti-TBEV Serum Level (VIEU/mL)		OR	95% CI	*p*
	≥165 *n* = 1996 (86.2%)	<165 *n* = 319 (13.8%)			
<1	512 (90.9)	51 (9.1)	reference group		
1	426 (88.9)	53 (11.1)	1.2	0.83–1.87	0.282
2	642 (82.4)	137 (17.6)	2.1	1.5–3.0	<0.001
3	205 (85.1)	36 (14.9)	1.7	1.1–2.8	0.015
≥4	211 (83.4)	42 (16.6)	2.0	1.3–3.1	0.002

OR—odds ratio; CI 95%—95% confidense intervals; VIEU—Vienna units; TBEV Tick—borne encephalitis virus. The model have been adjust for sex and age group <60 and ≥60 years.

**Table 5 medicina-56-00244-t005:** Cox regression analysis of risk factors related to seronegative values of anti-TBEV serum antibodies.

Variables	Anti-TBEV Serum Level (VIEU/mL)		RR	95% CI	*p*
	≥165 *n* = 1996 (86.2%)	<165 *n* = 319 (13.8%)			
Number of booster vaccination doses					
0	474 (75.7)	152 (24.3)	reference group		
1	873 (88.1)	118 (11.9)	0.14	0.11–0.18	0.001
2	378 (92.7)	30 (7.3)	0.06	0.04–0.09	0.009
3	145 (93.6)	10 (6.4)	0.02	0.01–0.05	<0.001
≥4	126 (93.3)	9 (6.7)	0.03	0.01–0.08	<0.001
Age group					
≥60 years	206 (84.4)	38 (15.6)	reference group		
<60 years	1790 (86.4)	281 (13.6)	0.77	0.60–0.98	0.040

RR—relative ratio; CI 95%—95% confidense intervals; VIEU—Vienna units; TBEV—Tick-borne encephalitis virus. The model have been adjust for sex.

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
