# Peer review of "Analysis of Multiple Risk Factors for Seronegative Rate of Anti-Tick-Borne Encephalitis Virus Immunization in Human Serum"

_medicina, 2020, doi:10.3390/medicina56050244_

Round 1

Reviewer 1 Report

This article dealt with vaccination against tick-borne encephalitis virus (TBEV) and concluded the personalization of vaccination schedule and the property of booster immunization.

I think this article will be potential important knowledge on further prevention of TBEV infection.

However, several obscure points are arisen.  Please address below indications.

  1. “line 215”, I could not find Table 4 in review manuscript. Please check results’ information.
  2. “Subsection of Results, 3.3 An influence of sex and age on anti-TEBV antibodies titer in serum”

The small number of female subjects was thought to be weaken the achievements and outcomes of this vaccine research.  Female alone data such as changes of anti-TEBV titer in basic vaccination or booster dose need be disclose even if these data correspond conclusion of the article.  Please embedded these data on the manuscripts or supplementary materials.  Followed by disclosure of female data, detailed contents of subsection 3.3 may be perspicuous.

Reviewer 2 Report

Analysis of multiple risk factors for negative and/or indeterminate results of anti-Tick-borne encephalitis virus antibodies in human serum

The finding of this work is merited to be published because it contribute to better understand risk factors of immunity persistence. Unfortunatelly, the manuscript requires major modification - verification of statistical analysis and structural rewriting.

Firstly general comments:

The abbreviation TBE or TBEV should be unified. It should be used corect abbreviation TBE or TBEV (not TEB or TEBV).

Abstract

Line 19: wining immunity - it should be: waning immunity or better immunity persistence.

Absent antibodies does not mean absence of the protection. The correlation between antibodies levels and protection exist only in antibodies measured by virus neutrlaisation titration.

Line 27: ... increased the odds ratio ... It is not correct, it should be ... increased a chance to lost anti-TBE antibodies  

Introduction

The section is very long and it does not focused on the study objectives. For example a description of disease or TBE virus etc. are not object of study.

Lines 95-96: … insufficient protective anti-TBE antibody level, considered as non-detectable or low antibodies status. Antibodies' concentrations under seropositive level could be still detectable. It should be reformulated.

Material and Methods

anti-TBE vaccination should be corrected: vaccination against TBE or TBE vaccination.

Table 1

There is enough to display the number of men - reader can calculate the number of women.

Furthemore, the table should be appropriately rearranged with 3 column, where one column will be for all participants and the other for those with antibodies level under seropositive limit.

It is not important to report how many participants were with no sufficient antibodies after primary of booster immunisation if their numbers are reported after the 1st, 2nd … dose.

Section 2.2. 

It should be focused on specificity of own method. If the serology assessment was performed according to manufacturer recommendation then this section is inappropriate. It is enough to report what kit was used. 

Section 2.3. 

It should be named such as Primary and booster immunisation. Text should be rewritten with no “-“.

Section 2.5. 

The definition of a seropositive rate or seronegative rate is missing. 

It should be reported abbreviation of 95% CI.

The antibodies' log-concentration passed the test of normality. Therefore, it could be used only one-way ANOVA test never Kruskal-Wallis test that is used for data non-normal distributed. It should be corrected.

When the multivariate (linear) regression and logistic regression were used? For what data. The results of multivariate linear regression are missing.

What confidence interval was included? One or two-sided? What level of significance was considered?

What groups or subgroups were compared (see line 140)?

What trend was evaluated (see line140)?

Results

Section 3.1.

Line 146: …frequency of non-positive results ...  it should be appropriately defined for example seronegative rate etc.

Table 2

It should be defined regular and irregular immunisation dependently of time since last immunisation. It should be appropriately evaluated seronegative rate in those with no sufficient antibodies dependently of the time since last vaccination with only one, two, three or ≥4 doses. Otherwise, there could be misinterpreting of results.

Section 3.2

Line 164: periodic booster should be replaced by for example regular booster dose etc.

Line 168: It should be defined abbreviation "y" as years.

It is missing the number of subjects complied the condition stated in name of this section. Therefore, the statement in lines 170-171 could be considered contrary to abstract outcome. 

Lines: 178-179: 

"With the passing of years, the number of subjects who need earlier vaccination reached 14.3%" - not understandable.

Line 180

Nobody with incomplete or irregular immunisation is considered non-responder. It is important to maintain the right terminology. Moreover, it was evaluated a persistence of immunity. If a participant had no sufficient antibodies after several months or years since the last vaccine reception it does not mean that he is a non-responder because the immediate immune response after vaccination was not evaluated in this study.

Section 3.3

Figure 3

It seems that lines of 95% CI for GMC are misplaced.

The basic immunisation means complete primary immunisation or any primary immunisation with one, two or three doses? Naturally, persistence of GMCs is dependent on the status of primary immunisation. It could be one of many reasons how to explain the age difference of persistent antibodies' GMCs.

Section 3.4.

The description of outcome does not correspond to RESULTs. It should be rewritten.

Line 213: ... received ≥4 periodic booster doses ...  It is not clear if subject received 3 primary doses plus 4 or more booster doses or if subject received 3 primary doses and 4th or other following dose, i.e. at least one booster dose. It should be appropriately written and defined in chapter "Materials and methods".

Discussion

The first two paragraphs are not suitable in section "Discussion". They are not linked with the study results.

The paragraph in lines 228-232 should be rewritten as limitation of study.

The waning immunity should not be evaluated in subject incompletely vaccinated, i.e. after the first or second dose of vaccine. 

A real immunity persistence in subjects with complete primary or booster immunisation should be recalculated. Furthermore, subjects with incomplete immunisation (<3 doses) could be assessed extra because it is obviously that they achieve higher seronegativity rate or lower seropositivity rate.

Reviewer 3 Report

In the paper "Analysis of multiple risk factors for negative and/or 2 indeterminate results of anti-Tick-borne encephalitis 3 virus antibodies in human serum" by Marta Janik et al, the authors try to identify risk factors for nonresponse to tick-borne encephalitis vaccine.

Major remarks

I believe that the results of this study do not support the conclusions, provided by the authors. Since only time elapsed since the last vaccination and the number of booster doses, but none of the individual-related factors (age, sex, potentially other covariates which were not included in the analysis) correlated with the probability of testing negative, individually based scheduling of booster doses does not seem to be judicious.

Minor remarks

1.Introduction

Better description of the clinical presentation of infection with TBE virus should be provided.

The reasoning for the present study should be given in the introduction section of the paper (discrepancies regarding immune response to TBE vaccine published in different studies). 

2. Materials and Methods

Table 1 should be moved to the Results section and on definition of study participants should be given in the Methods section.

The footnote a is not used in the Table 1. 

In Table 1, subheadings should be used. 

3. The paper needs extensive English language editing.

Author Response

Please see the atatachment

Reviewer 4 Report

The manuscript by Janik et al. analyzed several risk factors for negative and/or indeterminate results of anti-TEBV antibodies in human serum. Based on the results, the authors concluded that the vaccination schedule is better done in a personalized manner. Overall, this MS is straightforward and the data presented are convincible. This reviewer is supportive for publication and only have several minor suggestions.

1) Line 68-69, references should be added.

2) Basically, what is a better strategy for vaccination? How about the overall cost and is it practical? Some more discussions should be expanded.

Round 2

Reviewer 2 Report

The revision contributed to a substantial improvement of manuscript. I have the last remarks to odds ratio that should be presented as adjusted odds ratio (aOR) - see table 4 - because the logistic regression was used and I suppose that aORs are mutually adjusted with all predictors (variables).

In the table 4, there is unacceptable entry of aOR for variables - predictors: "Every 1 year from the last dose" and "Each subsequent dose of periodic booster" (in addition - incorrectly listed periodic booster instead of booster immunisation). It is necessary to select one reference value of each predictor and express aORs for each other value of the predictor related to the reference value of this predictor.

For example: If the reference value is considered 1 year after last vaccine dose, then cOR = 1 for this reference group and cOR = 0,80 (95% CI: 0,53-1,20) for <1 year after last dose; cOR = 1,72 (95% CI: 1,22-2,41) for 2 years, cOR = 1,41 (95% CI: 0,90-2,22) for 3 years etc. I could not calculate aOR because I had not entry data. Displayed data enabled to calculate only crude odds ratios (cOR). The results should be revised and rightly adapted. I am afraid that your current results could not to be correct.

One more thing. The number of booster doses is dependent on the the time of follow-up. Therefore this predictor is not suitable to assess with logistic regression that cannot reflect time-variable (variable correlated with time). For this purpose, there is more suitable the Cox regression. You should consider it. Otherwise your results will be burdened by the statistical bias.

Reviewer 3 Report

The authors provide data, that the currently used vaccination schedules are insufficient for a large number of people (without providing individual characteristics, beyond the common ones, i.e. time since the last vaccination and the number of booster doses). Therefore I propose to somewhat soften the conclusion that the vaccination scheme should be personalized, and rather propose that they should be reevaluated/rescheduled.

Round 3

Reviewer 2 Report

The revision contributed to a substantial improvement of manuscript. I am agree to publish it.

Reviewer 3 Report

I find the manuscript exceptable for publication.